# Knowledge, attitudes, and practices of face mask utilization and associated factors in COVID-19 pandemic among Wachemo University Students, Southern Ethiopia: A cross-sectional study

**Yilma Markos Larebo**[1]*, **Desta Erkalo Abame**[2]

**1** Department of Epidemiology, College of Medicine and Health Sciences, Wachemo University, Hossana, Ethiopia, **2** Department of Reproductive Health, College of Medicine and Health Sciences, Wachemo University, Hossana, Ethiopia

* yilmamark@gmail.com

**Data Availability Statement:** All relevant data are within the paper and its Supporting information files.

## Abstract

### Introduction

The widespread use of face masks by the general public may help to prevent the spread of viruses. Face masks are thought to be a good strategy to protect against respiratory diseases such as the Coronavirus. Identifying student knowledge, attitude, and practice about the use of face masks is crucial to detect vulnerabilities and respond rapidly to avoid the spread of the infection. This study aimed to determine the knowledge, attitude, and practices of face mask utilization and associated factors in the COVID-19 pandemic among college students.

### Methods

A cross-sectional study was performed from February to March 2021 among 764 students from Wachemo University, Southern Ethiopia. A multistage sampling technique was used in the study. The sample size for each department was allocated in proportion to the number of students in that department, and each respondent was chosen using a simple random sampling procedure. Data were collected using a pre-tested self-administered questionnaire and analyzed using SPSS version 26. To predict the relationship between the predictor and outcome variables, a logistic regression model was used. At a p-value of 0.05, statistical significance was declared.

### Results

The study showed that the overall knowledge of the students was 223 (29.2%), their attitude was 673 (88.1%), and their practice was 684 (89.5%). The students from the college natural and computational sciences (AOR: 0.23; 95%CI: 0.13, 0.40) and students having good

**Funding:** This study was sponsored by the Wachemo University (WCU), as one of the 3rd generations higher Institution University, Hossana, Ethiopia, as part of the community research submitted to research and community service. The funder had no role in the design of the study, data collection, and analysis, interpretation of the data, and preparation of the manuscript.

**Competing interests:** The authors declared that there is no conflict of interest.

knowledge (AOR = 4.40; 95%CI; 2.13, 9.14) were found to be independently associated with face mask utilization.

## Conclusion

When compared to other researches, the knowledge about the usage of face masks in this study was low, but the attitudes and practices were high. Authorities in areas that are in danger of a COVID-19 pandemic should plan and implement public awareness and education initiatives.

## 1. Introduction

Coronavirus disease (COVID-19) appeared as a new public health issue in Wuhan, China, with the public health problem that started in bats also spread to people via unknown intermediary species at the end of 2019 [1, 2]. High-grade fever, cough, sore throat, dyspnea, tiredness, and malaise were frequently observed symptoms in patients infected with the COVID-19 virus. The weaker immune systems of chronic disease patients and the elderly make the effects of COVID-19 more potent to them, which could result in death for those suffering from these comorbidities [1].

The disease has already spread to 219 countries around the world, prompting the World Health Organization (WHO) to declare it a pandemic. As of March 2021, there had been more than 120 million positive COVID-19 cases reported worldwide, with more than 2.5 million deaths and approximately 4 million vaccination doses provided [1, 3, 4].

Severe Acute Respiratory Syndrome Coronavirus 2 (SARS-CoV-2) is a virus that can infect persons of all ages and from all walks of life. It indiscriminately kills as it can affect healthy adults and people suffering from health problems, and it spreads swiftly at an exponential pace [1, 2]. The virus can be spread in two ways, namely, inhalation and contact with contaminated droplets. The incubation period for the disease varies from 2 to 14 days [1].

The disease could have major consequences in both developed and poor nations, where healthcare facilities and resources are inadequate; nevertheless, face coverings help to restrict the proliferation of COVID-19 [1, 5–7]. Preventive measures, which include the use of masks, hand hygiene, a physical distance of at least 1 meter, proper ventilation in indoor environments, monitoring, contact tracing, quarantine, isolation, and other infection prevention and control (IPC) measures, are all necessary to prevent SARS-CoV-2 transmission from person to person [7–9].

In collaboration with the government, health officials recommended that the public use face masks or coverings to reduce the risk of transmission, with authorities requiring their use in specific environments, such as public transportation, stores, schools (high, colleges, and universities), police stations, and other public places. Health officials have recommended that medical-grade face masks, such as respirators, be prioritized for use by medical personnel to prevent the shortage of supply for this product. Meanwhile, cotton masks are recommended for the public [8, 10, 11].

During the early stages of the pandemic, public health messages about wearing masks were inconsistent, conflicting, and tinged with contemptuous comments, which caused public discontent and even confusion. Eventually, the recommendations have evolved in lockstep with scientific understanding [8].

The widespread usage of face masks can help reduce virus transmission between individuals who are infected with the virus but have not yet acquired symptoms, as well as between individuals who do not have symptoms but are infected with the virus [1]. Ethiopia has substantially reversed course, aiming to achieve national and state records for new COVID-19 cases by September 2020. As of March 2021, Ethiopia had reported 181,869 confirmed cases and 2,602 deaths [3].

The World Health Organization has recommended that healthy persons wear masks indoors and outdoors for all students, employees, and tourists, with valid medical exemptions, when two meters of social distance cannot be maintained, since June 2020. In situations when physical isolation is problematic, non-medical masks are used to track the transmission of COVID-19 [3, 11, 12]. Many schools and universities require students to wear masks in public places and when they are within six feet of others [13].

Even if the study location is known to the investigators, there is little information about knowledge, attitudes, and practices face mask utilization among university students in Ethiopia. Therefore, this study was aimed to determine the knowledge, attitudes, and practices of face mask utilization, as well as associated factors, among Wachemo University (WCU) students in southern Ethiopia; additionally, the findings will be used as a guideline for those interested in researching related topics.

## 2. Methods and materials

### 2.1 Study setting, design, and period

An institution-based cross-sectional study design was conducted among Wachemo University's main campus regular students from February to March 2021 who are recruited from the different departments. Wachemo University is of higher public education institution established in 2012 in Hossana town located in Southern Region at 232km south of Addis Ababa, which is the capital city of Ethiopia, and 194km far from the regional city Hawassa. WCU comprises of Durame campus and Nigist Eleni Mohamed Memorial Comprehensive Specialized Hospital (NEMMCSH) in addition to the Main Campus (MC). There are six colleges in WCU-Main-Campus and 10,371students enrolled in the regular program and from those 3,691 are female and 6,680 are male in the 2020/2021 academic year in the University [14–16].

### 2.2 Sample size determination

The sample size was calculated using the single population proportion formula, considering the following assumptions and taking prevalence of 45.3% which is a study conducted in Police Health Facilities of Addis Ababa, Ethiopia [5].

$$n = \frac{(Z\alpha/2)^2 p(1-p)}{d^2} \tag{1}$$

Where n = the desired sample size P = attitude of the healthcare provider towards proper face mask utilization = 45.3% (which is taken from attitude related study conducted at Addis Abebe police health facilities, Ethiopia,2020) Z1-α/2 = Critical value at 95% confidence level (1.96) d = the margin of error = 5% n = $(1.96)^2 0.453(1-0.453)$ / $(0.05)^2$ = 381 and Using design effect (Deff = 2), because two stage sampling technique used, the final sample size required is: 2*381 = 762, for possible none response during the study the final sample size was increased by 5% to: n = 762 +5% of 762 which is: 38.1 By adding then, the total sample sizes was 800.

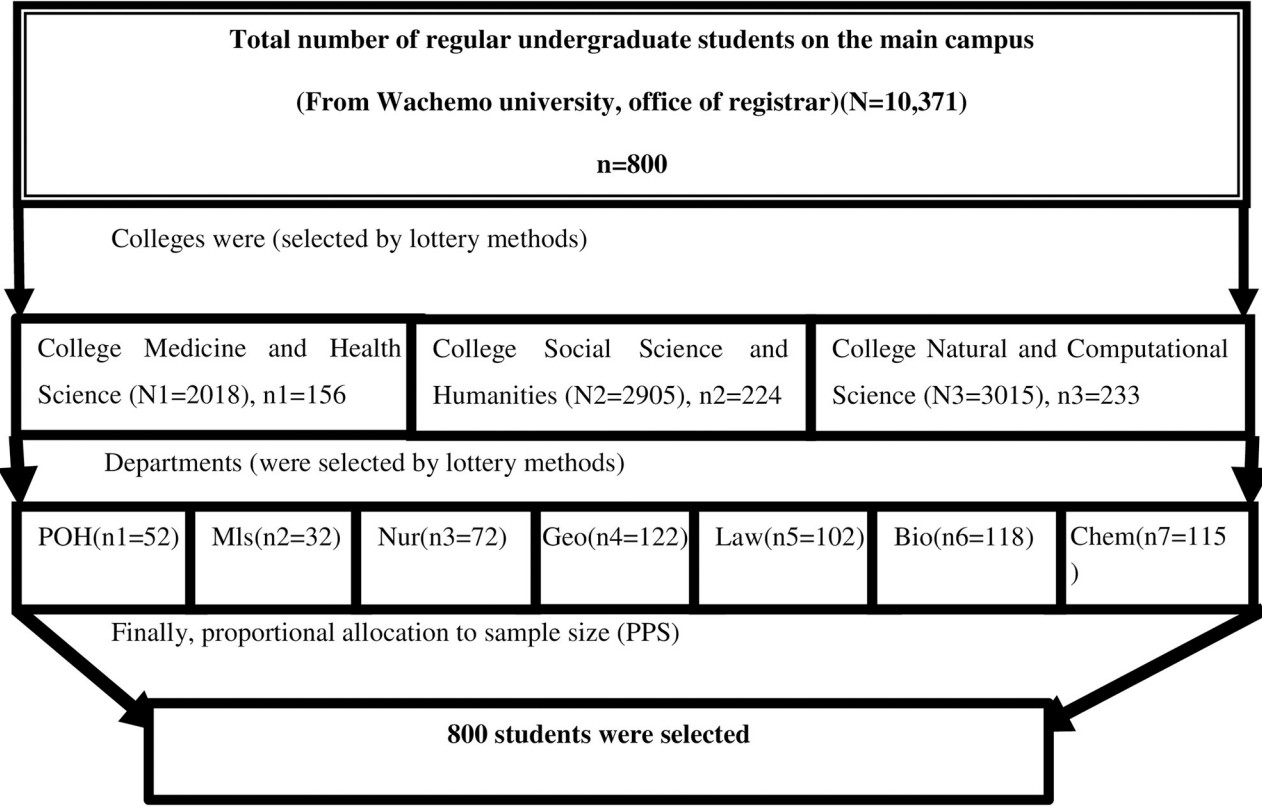

**Fig 1. Schematic presentation of sampling procedure respondents on about knowledge, attitudes, and practices of face mask utilization and associated factors in COVID-19 pandemic among Wachemo university students, southern Ethiopia: February to March 2021 (n = 800).** Where: POH = department public health officer, Mls = department of medical laboratory, Nur = department of nursing, Geo = department of geography, Law = department of law, Bio = department of biology and Chem = department of chemistry.

## 2.3 Sampling procedure

All regular undergraduate students registered for the 2020/2021 academic year in WCU main campus, were considered as the source population. A multistage sampling technique was used in the investigation. The first stage was created utilizing colleges, with three out of six colleges being chosen by lottery to increase representativeness. Seven departments at the selected colleges were proportionally allocated in the second step. The proportional sample size for each department was allocated in proportion to the number of students per department, and each respondent was chosen using a simple random sampling procedure. All regular undergraduate students registered from 1st to 6th year and attending their education during the study period in the main campus, WCU were included in the study. All weekend undergraduate students registered from 1st to 6th year and attending their education during the study period in the main campus were excluded from the study (Fig 1).

## 2.4 Measurement of variables

The purpose of this study was to determine WCU students' knowledge, attitude, and practices regarding face mask utilization and associated factors in the COVID-19 pandemic. The primary dependent variable in this study was practice face mask utilization, but knowledge and attitude toward face mask utilization were also considered secondary dependent variables. The

predictor variables were respondents' basic socio-demographic characteristics such as gender, age, religion, family's main source of income, marital status, students per dorm, current living situation, ethnicity, residency before joining the university, college, department, study years, cumulative grade point average (CGPA), monthly income, and sources of information about cloth face coverings.

In this study, knowledge about face masks and they are used were computed as follows: each correct response in the knowledge category was scored 1 and each incorrect response was scored 0. The final score was calculated and then labeled based on a score out of 9. The correct response of >7 out of 9 questions (>80%) was considered as good knowledge and ≤7 (≤80%) was considered as poor knowledge [5].

The attitude towards face mask utilization was measured by asking eight questions (e.g. willing to know the correct steps of wearing a face mask) to describe their level of agreement in a 5-point Likert Scale response options, scored from 1 to 5, strongly disagree, disagree, neutral, agree, and strongly agree. Subscale scores were obtained by summing item scores and dividing them by the total number of items. If it is above or equal to the average it was considered a positive attitude [5].

The practice of face mask utilization was analyzed as follows: each correct response in the practice category was scored 1, and each incorrect response was scored 0. The correct response of > 14 out of 18 questions (>80%) was considered as good practice and ≤ 14 (≤80%) was considered as poor practice [5].

## 2.5 Data collection and quality assurance procedures

The questionnaire was developed by reviewing different literature on the utilization of face masks and the guidelines of the center for health protection World Health Organization (WHO) and the Communicable Disease Control (CDC) [17–19] and in consultation with experts from different fields to check the relevance and make necessary changes according to the study requirements. The questions were modified according to the suggestions received from the committee and output from the pre-test. Guidelines for layout, question design, formatting, and pretesting were followed. The questionnaire developed by the investigators contained the following 4 sections: 1 basic demographic characteristics (age, sex, religion, source of income, marital status, students per dorm, living situation, ethnicity, residence before joining university, college, department, study years, cumulative grade point average (cGPA), source of information and monthly income in birr), and 2 knowledge, 3 attitude, and 4 practices regarding face mask utilization. Three days of training were provided for data collected using a self-administered method by two trained data collectors and one supervisor. Data collectors pre-tested the questionnaire on 40 students from the Durame Campus of WCU branches two weeks before the primary study's data collection began. The collected data were checked for completeness, accuracy, clarity, and consistency by the principal investigator. The questionnaire was prepared in English and then translated common language Amharic to check the consistency of the items and back to English to verify the accuracy in the common language Amharic. Face validation of the questionnaire was determined.

## 2.6 Data processing and analysis

The coded data were entered in Epi Data version 3.1 and exported for analysis to SPSS version 26. The key investigator was responsible for data entry. The descriptive analysis was performed and presented in frequency, using tables, graphs, and charts. In the bivariate analysis, variables having P-values less than 0.25 were entered into the multivariate analysis.

Bivariate analysis was performed to select variables for multivariate analysis. But, statistical significance was tested at the level of 5% at the multivariate level. Adjusted odds ratios (AOR) and 95% confidence interval using logistic regression were used to verify the existence and intensity of the correlation between independent and dependent variables. The fitness of the model was tested using the 0.796 Hosmer and Lemeshow test.

## 2.7 Ethical approval

The ethical approval was obtained from the ethical review committee (ERC) of the WCU College of Medicine and Health Sciences. Before beginning the investigation, permission was obtained from universities, colleges, and departments. Respondents were given written consent about the study, its goals, effects, and the significance of the data before they were enrolled. To ensure confidentiality, all information was rendered anonymous.

# 3. Results

## 3.1 Characteristics of respondents

Out of the 800 undergraduate students who were eligible to participate in the study, 36 were excluded (36 data were incomplete and were not considered for analysis), resulting in a 95.5% response rate. Out of 764 students included in the study, 522(68.3%) of the respondents were male with almost half 378(49.5%) of them in the age category 20–24 years. The mean age of students was 24.49 with (SD) ± (1.96) years and almost half proportions 387(50.7%) of the respondents were protestant. More than half of students 405(53%) the family's main sources of income were from the government employee and the majority 553(72.4%) were single. 291 (38.1%) of respondent students were from Hadiya ethnicity. Most respondents 707(92.5%) were from an urban area and the majority of respondents 516(87.5%), 2 to 4 students per dorm and were currently living in a dormitory 632(82.7%). The 337(44.1%) of the students were from medicine and Health Sciences College, and nearly half 367(48%) students were a class year of four. All students reported their cumulative Grade Point Average (cGPA); a large proportion of the 464(60.7%) had a cGPA between 2.00–3.49 (good) points. Less than one-third 205(26.8%) of students received some monthly pocket money between 300–499 Ethiopian Birr (ETB) (Table 1).

Almost all 752(98.4%) and 687(89.9%) of the students know surgical masks can protect from COVID- 19 and correct use of surgical face masks, respectively. 482 (63.1%) and 356 (46.6%) of the respondents know the layers of the surgical mask and the layer which acts as a filter media, respectively. Concerning the type of mask, for protection against COVID-19, 347 (45.4%) of the respondents were responded to the correct answer. All most third 272(35.6%) of them know the duration of surgical mask use (Table 2).

This study identified majority of 599 (78.4%) and 660 (86.4%) of the college students were willing to know the correct steps of face mask-wearing and believe that face masks should be carefully put on and taken off, respectively. Regarding the effectiveness of face masks in preventing the spread of droplets, 668 (87.4%) of them believe it is effective while 91 (11.9%) of them disagreed. Half 639 (83.6%) of the professionals believe in changing face masks before going to another patient while 112 (14.65%) think it is not necessary to change face masks before going to another patient. Close to one-fourth of 335 (43.8%) disagreed it is not necessary to wear a face mask while in contact with patients and 320 (41.8%) said it is necessary to wear a face mask as am afraid of getting COVID-19. Almost 113 (14.79%) of the professionals believe that it is necessary to wear a face mask as being infected with COVID-19 is the worst thing that could happen to me (Table 3).

**Table 1. Sociodemographic characteristics of the respondents on knowledge, attitudes, and practices of face mask utilization and associated factors in COVID-19 pandemic among Wachemo university students, southern Ethiopia: February to March 2021 (n = 764).**

| Variables | Categories | n (%) |
|---|---|---|
| Sex | Male | 522(68.3) |
| | Female | 242(31.7) |
| Age in years | ≤ 20 | 207(27.1) |
| | 20 to 24 | 378(49.5) |
| | ≥25 | 179(23.4) |
| Religion | Orthodox | 253(33.1) |
| | Muslim | 35(4.6) |
| | Protestant | 387(50.7) |
| | catholic | 36(4.7) |
| | others* | 53(9.9) |
| Family's main source of income | Agriculture | 103(13.5) |
| | Government job | 405(53) |
| | Trade | 190(24.9) |
| | NGO/private firm work | 66(8.6) |
| What is your marital status now | Single | 553(72.4) |
| | Married | 211(27.6) |
| Number of students per dorm | ≤2 | 190(24.9) |
| | 2 to 4 | 516(67.5) |
| | ≥5 | 58(7.6) |
| Current living situation | Dormitory | 632(82.7) |
| | Rented accommodation | 103(13.5) |
| | Others** | 29(3.8) |
| Ethnicity | Hadiya | 291(38.1) |
| | Kembata | 214(28) |
| | Amhara | 53(6.9) |
| | Tigre | 16(2.1) |
| | Gurage | 32(4.2) |
| | Oromo | 63(8.2) |
| | Wolaita | 54(7.1) |
| | others*** | 41(5.4) |
| Residence before joining the university | Urban | 707(92.5) |
| | Rural | 57(7.5) |
| College | Medicine and health sciences | 337(44.1) |
| | Natural and computational sciences | 219(28.7) |
| | College social science and humanities | 208(27,2) |
| Department | Public health officer | 120(15.7) |
| | Nursing | 116(15.2) |
| | Medical laboratory | 101(13.2) |
| | Biology | 115(15.1) |
| | Chemistry | 104(13.6) |
| | Geography | 107(14) |
| | Law | 101(13.2) |
| Study years | First Year | 32(4.2) |
| | Second Year | 248(32.5) |
| | Third Year | 117(15.3) |
| | Fourth Year and Above | 367(48) |

(*Continued*)

**Table 1.** (Continued)

| Variables | Categories | n (%) |
|---|---|---|
| Cumulative grade point average (cGPA) | 3.5–4.00(Excellent) | 232(30.4) |
| | 2.00–3.49(Good) | 464(60.7) |
| | <2.00(Poor) | 68(8.9) |
| What is your monthly income in birr(average pocket money per month) | 1–100 | 115(15.1) |
| | 101–299 | 93(12.2) |
| | 300–499 | 205(26.8) |
| | 500–999 | 198(25.9) |
| | 1000 and above | 153(20) |
| Sources of information about cloth face coverings | Internet | 299(39.1) |
| | TV | 145(19) |
| | Social media | 86(11.3) |
| | E-mail message | 45(5.9) |
| | Newspapers | 79(10.3) |
| | Grocery store | 25(3.3) |
| | Radio | 85(11.1) |

Where: TV;Televesion * = 23 Joba witness, 6 No religion, 24 Hawariyat, ** = 19 Living with relatives, 10 family members, *** = 8 Alaba, 9 Sidama, 5 Kaffa, 7 Dawro, or 12 Somali. Of the 764 study respondents, the majority, 541 (70.8%) with 95% CI [67.4, 73.8] had a poor knowledge with a mean of 0.292 and standard deviation of ±0.455 (Fig 2)

Almost more than one-third 275 (36%) of the students removed their face mask if there is a need to talk to the patient while 578 (75.7%) of the store using a mask in a bag for later use if not sick. 743 (97.3%) of the study respondents do wear a face mask in public places and most 743 (97.3%) of them wore face masks on hospital premises. A significant number of 743

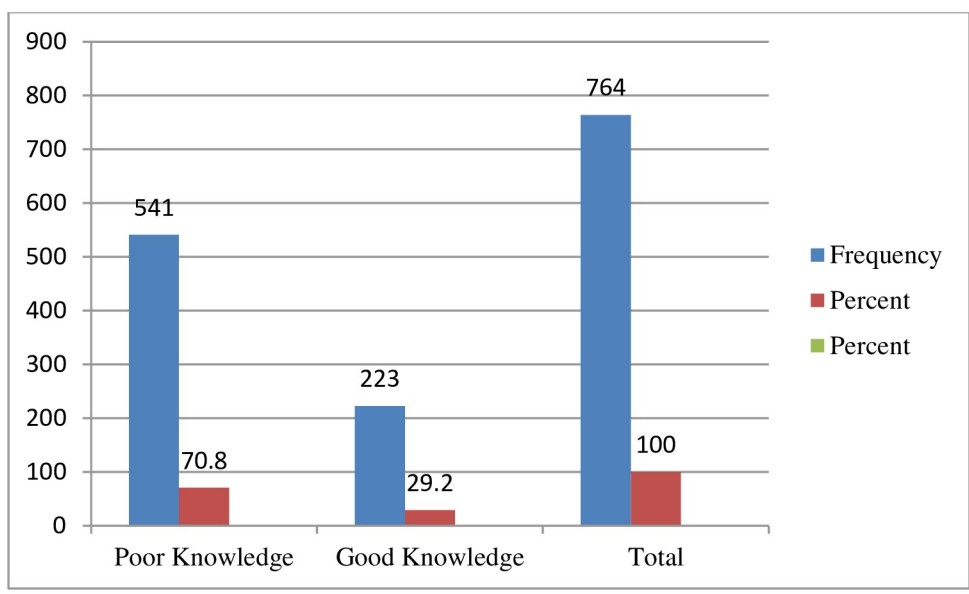

**Fig 2. Knowledge-related questioner respondents on about knowledge, attitudes, and practices of face mask utilization and associated factors in COVID-19 pandemic among Wachemo university students, southern Ethiopia: February to March 2021 (n = 764).**

**Table 2. Knowledge-related questioner on knowledge, attitudes, and practices of face mask utilization and associated factors in COVID-19 pandemic among Wachemo university students, southern Ethiopia: February to March 2021 (n = 764).**

| Variables | Categories | n (%) |
|---|---|---|
| Can wearing a surgical mask protect you from COVID-19 | Yes | 752(98.4) |
| | No | 12(1.6) |
| Which is the correct way of using a surgical face mask to protect against COVID-19 | The white side facing in (correct) | 687 (289.9) |
| | The white side facing out | 77(10.1) |
| How many layers are there in a surgical mask | Two | 482(63.1) |
| | Three | 277(36.3) |
| | Four | 5(0.7) |
| Which layer acts as a filter media barrier from covid 19 virus | First | 356(46.6) |
| | Middle(correct) | 223(29.2) |
| | Last | 185(24.2) |
| Which type of masks protect against COVID-19 | 99% BFE&PFE | 347(45.4) |
| | 97% BFE&PFE | 45(5.9) |
| | 95% BFE&PFE(correct) | 177(23.2) |
| | 91% BFE&PFE | 195(25.5) |
| How long can you wear a surgical mask | 8 hours(correct) | 272(35.6) |
| | 4hours | 86(11.3) |
| | 2 hours | 406(53.1) |
| For proper wearing, to which extent the surgical mask should cover | the mouth only | 118(15.4) |
| | mouth and nose | 293(38.4) |
| | Nose, mouth, and chain (correct) | 353(46.2) |
| What is the purpose of the metal strip on a surgical mask | No purpose | 207(21.7) |
| | To fit on the nose(correct) | 453(59.3) |
| | To fit on the chin | 145(19) |
| Is the cloth facial mask as effective as a regular surgical facial mask | Yes | 214(28) |
| | No | 550(72) |

Where: BFE, bacterial filtration efficiency; PFE, particle filtration efficiency.

Out of 764 total respondents, 673 (88.1%) had positive attitudes with a 95% CI [85.9, 90.4] and a mean of 0.881 and a standard deviation of ± 0.324 (Fig 3).

(97.3%) of students do clean their hands before wearing their face mask and 743(97.3%) of them check the inside and outside of the mask before wearing it. 743(97.3%) of them did not clean their hands after taking off the mask and 314 (77.0%) re-used a single-use mask (Table 4).

## 3.2 Overall utilization face mask

The study showed that the overall knowledge of the students was 223 (29.2%), their attitude was 673 (88.1%), and their practice was 684 (89.5%) (Fig 5).

**3.2.1 Factors affecting knowledge about face mask utilization.** Multivariable analysis was used to control potential confounders. Having students per dorm ≤2 and ≥5 in number, rural residency, college natural, and computational sciences, social sciences and law, grade point <2.00(poor), and having poor practices were found to be significantly associated with knowledge about face masks utilization with a P-value < 0.05.

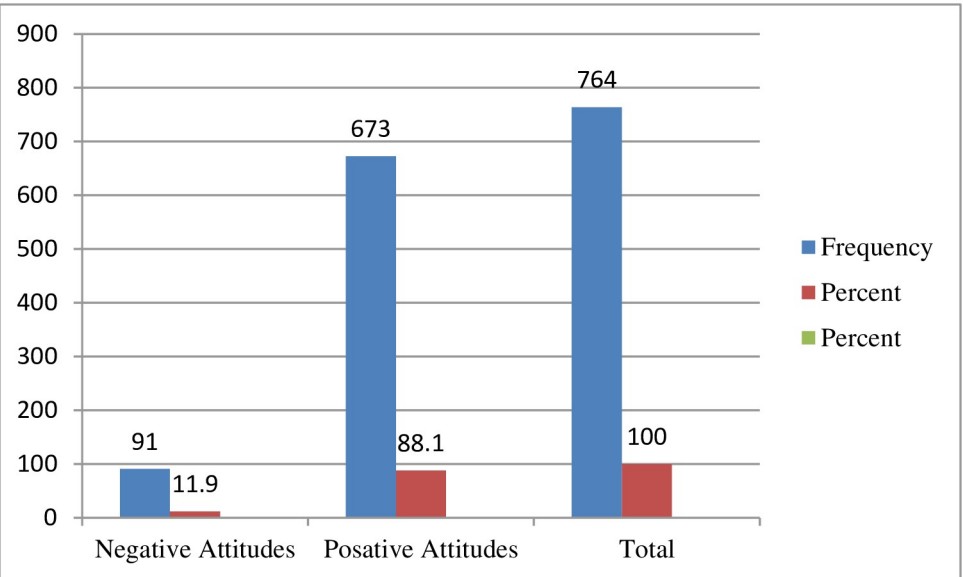

**Fig 3. Attitude-related questioner respondents on about knowledge, attitudes, and practices of face mask utilization and associated factors in COVID-19 pandemic among Wachemo university students, southern Ethiopia: February to March 2021 (n = 764).**

In this study, having students per dorm ≤2 in number were 90.4% less likely and having students per dorm ≥5 in number were 3.87 times more likely to have poor knowledge than students per dorm 2 to 4 number(AOR: 0.096; 95%CI: 0.025, 0.362) and (AOR: 3.861; 95%CI: 1.595, 9.344) in face mask utilization respectively, students having rural residence was 68.9% less likely have poor knowledge in face mask utilization in COVID 19 pandemic when compare with urban residences, students from college of natural and computational sciences were 6 times and students from college of social sciences and law were 2.76 times more likely poor knowledge than on utilization of face mask when compare to students from college of medicine and health sciences (AOR: 6.102;95%CI: 3.497,10.647) and (AOR: 2.759;95%CI:

**Table 3. Attitude-related questioner on knowledge, attitudes, and practices of face mask utilization and associated factors in COVID-19 pandemic among Wachemo university students, southern Ethiopia: February to March 2021 (n = 764).**

| Variables | Strongly Disagree | | Disagree | | Neutral | | Agree | | Strongly Agree | |
|---|---|---|---|---|---|---|---|---|---|---|
| | n | % | n | % | n | % | n | % | n | % |
| Willing to know the correct steps of wearing a face mask | 9 | 1.2 | 91 | 11.9 | 65 | 8.5 | 119 | 15.6 | 480 | 62.8 |
| Needs to be carefully put on and taken off | 5 | 0.7 | 86 | 11.3 | 13 | 1.7 | 289 | 37.8 | 371 | 48.6 |
| Keeps individual from touching mucous members | 6 | 0.8 | 94 | 12.3 | 4 | 0.5 | 285 | 37.3 | 375 | 49.1 |
| Very effective at preventing infectious droplets from spreading | 59 | 7.7 | 32 | 4.2 | 5 | 0.7 | 342 | 44.8 | 326 | 42.7 |
| Necessary to change the face mask before going to another patient | 59 | 7.7 | 53 | 6.9 | 13 | 1.7 | 269 | 35.2 | 370 | 48.4 |
| Necessary to wear a face mask when in contact with patients. | 190 | 24.9 | 145 | 19 | 12 | 1.6 | 235 | 30.8 | 182 | 23.8 |
| Necessary to wear a face mask as am afraid of getting COVID-19 | 202 | 26.4 | 230 | 30.1 | 12 | 1.6 | 194 | 25.4 | 126 | 16.5 |
| Necessary to wear a face mask as infected with COVID-19 is the worst thing that could happen to me | 364 | 47.6 | 228 | 29.8 | 59 | 7.7 | 100 | 13.1 | 13 | 1.7 |

Out of 764 total respondents, almost more than half of 89.5% respondents with good practices with 95% CI [87.3, 91.9] with a mean of 0.89 and standard deviation of ±0.331 (Fig 4).

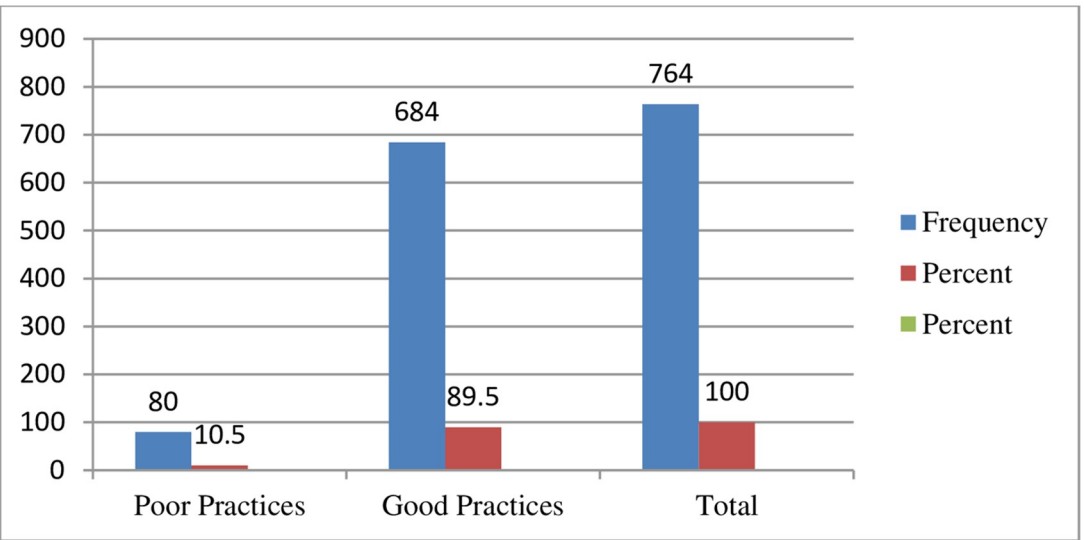

**Fig 4. Practices-related questioner respondents on about knowledge, attitudes, and practices of face mask utilization and associated factors in COVID-19 pandemic among Wachemo university students, southern Ethiopia: February to March 2021 (n = 764).**

1.461,5.208) respectively, those students having cumulative grade point <2.00(poor) were 3.5 times more likely poor knowledge than having students cumulative grade point between 2.00–3.49(good) and those students having poor practices were 71.4% times less likely to have poor knowledge on face mask utilization than having students with history of good practices (Table 5).

### 3.3 Factors affecting the attitude towards face mask utilization

Attitude towards face mask utilization was significantly associated with the students with history rented accommodation, with rural residences, college of social sciences and law, and students having good knowledge were found to be significantly associated with attitudes about face mask used with a P-value < 0.05.

In this study, those students having a history of rented accommodation were 76.5% times less likely to have negative attitudes than on face mask utilization than those students were having living in the dorm (AOR: 0.235;95%CI: 0.093,0.590), students from rural residences were 94.1 times less likely to have negative attitudes than students from urban residences (AOR:0.059;95% CI;0.025,0.138), having a college of social sciences and law (AOR:0.079;95% CI;0.034,0.184) and students having good knowledge were 4.56 times more likely to have negative attitudes when compared students having poor knowledge on face mask utilization (AOR:4.556;95% CI;2.130,9.740) (Table 6).

### 3.4 Factors affecting the practice of face mask utilization

College of natural and computational science, with having good knowledge about face mask utilization was significantly associated with the practice of face mask utilization.

The students from natural and computational sciences were 76.8% less likely associated with poor practices in the utilization face mask in COVID -19 pandemic when compared to the college medicine and health sciences (AOR: 0.23; 95%CI: 0.13, 0.40) and students having good knowledge were 4.4 times more likely to had poor practices on face mask utilization than

**Table 4. Practices-related questioner on knowledge, attitudes, and practices of face mask utilization and associated factors in COVID-19 pandemic among Wachemo university students, southern Ethiopia: February to March 2021 (n = 764).**

| Variables | Categories | n (%) |
|---|---|---|
| Remove his/her mask if there is a need to talk to pt. | Yes | 275(36) |
| | No | 489(64) |
| Sore used a mask in a bag for later use if not sick | Yes | 578(75.7) |
| | No | 186(24.3) |
| Wearing a mask in public places | Yes | 734(96.1) |
| | No | 30(3.9) |
| Wear a mask on hospital premises | Yes | 743(97.3) |
| | No | 21(2.7) |
| Before doing a mask, clean their hands | Yes | 743(97.3) |
| | No | 21(2.7) |
| Before wearing the mask identified the inside and outside mask | Yes | 743(97.3) |
| | No | 21(2.7) |
| Confirm the metal noseband on the top | Yes | 743(97.3) |
| | No | 21(2.7) |
| Place the loop around the ear | Yes | 743(97.3) |
| | No | 21(2.7) |
| Pull the top and bottom of the mask to extend the folds | Yes | 680(89.5) |
| | No | 80(10.5) |
| Press the noseband | Yes | 743(97.3) |
| | No | 21(2.7) |
| Do not touch the mask | Yes | 743(97.3) |
| | No | 21(2.7) |
| Do not eat drink/smoke while wearing the mask | Yes | 680(89.5) |
| | No | 80(10.5) |
| Remove the mask from the face touching only the bands | Yes | 680(89.5) |
| | No | 80(10.5) |
| Avoid pulling the mask up over my forehead or down over my chin | Yes | 743(97.3) |
| | No | 21(2.7) |
| Dispose of the mask when soiled/wet | Yes | 680(89.5) |
| | No | 80(10.5) |
| Clean hands after taking off | Yes | 743(97.3) |
| | No | 21(2.7) |
| Not reuse the single-use mask | Yes | 743(97.3) |
| | No | 21(2.7) |
| Everyone needs to wear a cloth face covering when they are out in public | Yes | 725(94.9) |
| | No | 39(5.1) |

students having poor knowledge about face mask utilization (AOR = 4.40; 95%CI;2.13,9.14) were found to be independently associated practices of face mask utilization(Table 7).

## 4. Discussion

The major purpose of this study was to assess the knowledge, attitudes, and practices of face mask utilization, as well as associated factors, among students at WCU in Southern Ethiopia during the COVID -19 pandemic. Masks are part of a larger package of preventative and control strategies that can help to restrict the spread of respiratory viral infections like COVID-19.

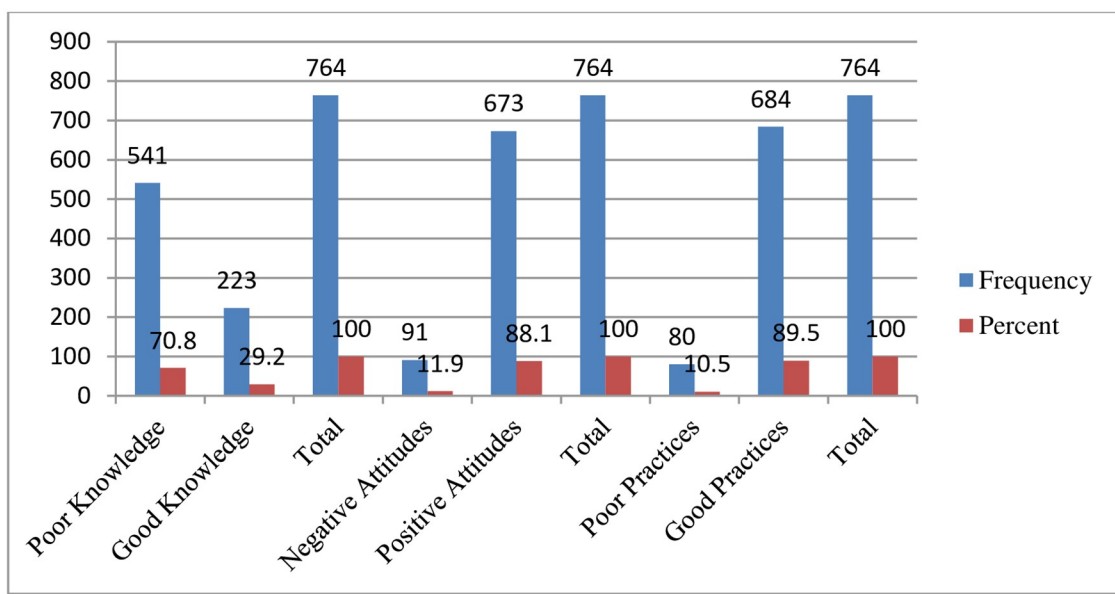

**Fig 5. The overall face mask utilization and associated factors respondents on about knowledge, attitudes, and practices of face mask utilization and associated factors in COVID-19 pandemic among Wachemo university students, southern Ethiopia: February to March 2021 (n = 764).**

**Table 5. Factors affecting knowledge of the respondents on knowledge, attitudes, and practices of face mask utilization and associated factors in COVID-19 pandemic among Wachemo university students, southern Ethiopia: February to March 2021 (n = 764).**

| Variable | Knowledge Of Face Mask | | | | |
|---|---|---|---|---|---|
| | Poor | Good | COR(95%CI) | AOR(95%CI) | P-value |
| Students per dorm | | | | | |
| ≤2 | 186(34.4) | 4(1.8) | 0.031(0.011,0.086) | 0.096(0.025,0.362)* | 0.001 |
| 2 to 4 | 306(56.6) | 210(94.2) | 1 | 1 | |
| ≥5 | 49(9.1) | 9(4) | 0.268(0.129,0.557) | 3.861(1.595,9.344)* | 0.003 |
| Residency | | | | | |
| Urban | 493(91.1) | 214(96) | 1 | 1 | |
| Rural | 48(8.9) | 9(4) | 0.432(0.288,0.896) | 0.311(0.129,0.750)* | 0.009 |
| College | | | | | |
| Medicine and health sciences | 207(38.3) | 130(58.3) | 1 | 1 | |
| Natural and computational sciences | 150(27.7) | 69(30.9) | 0.732(0.511,1.050) | 6.102(3.497,10.647)* | 0.001 |
| Social sciences and law | 184(34) | 24(10.8) | 0.208(0.129,0.335) | 2.759(1.461,5.208)* | 0.002 |
| Cumulative grade point average (cGPA) | | | | | |
| 3.5–4.00(Excellent) | 150(27.7) | 82(36.8) | 2.437(1.705,3.485) | 1.142(0.752,1.733) | 0.535 |
| 2.00–3.49(Good) | 379(70.1) | 85(38.1) | 1 | 1 | |
| <2.00(Poor) | 12(2.2) | 56(26.1) | 2.808(1.686,4.516) | 3.527(1.084,8.677*) | 0.001 |
| Practices | | | | | |
| Poor | 71(13.1) | 9(4) | 0.278(0.137,0.567) | 0.286(0.107,0.762)* | 0.012 |
| Good | 470(86.9) | 214(96) | 1 | 1 | |

Where 1 = Reference,* shows the variable significant at p-value < 0.05 in multi variable analysis.

**Table 6. Factors affecting attitude of the respondents on knowledge, attitudes, and practices of face mask utilization and associated factors in COVID-19 pandemic among Wachemo university students, southern Ethiopia: February to March 2021 (n = 764).**

| Variable | Attitude Of Face Mask | | | | |
|---|---|---|---|---|---|
| | Negative | Positive | COR(95%CI) | AOR(95%CI) | P-value |
| Current living situation | | | | | |
| Dormitory | 57(33.5) | 575(85.4) | 1 | 1 | |
| Rented accommodation | 32(35.2) | 71(10.5) | 0.220(0.134,0.362) | 0.235(0.093,0.590)* | 0.002 |
| Others | 2(2.2) | 27(4) | 1.338(0.310,5.773) | 0.278(0.048,1.619) | 0.155 |
| Residency | | | | | |
| Urban | 59(64.8) | 648(96.3) | 1 | 1 | |
| Rural | 32(35.2) | 25(3.7) | 0.071(0.40,0.128) | 0.059(0.025,0.138)* | 0.001 |
| College | | | | | |
| Medicine and health sciences | 30(33) | 307(45.6) | 1 | 1 | |
| Natural and computational sciences | 50(54.9) | 169(25.1) | 0.330(0.202,0.539) | 1.182(0.472,2.958) | 0.721 |
| Social sciences and law | 11(12.1) | 197(29.3) | 1.750(0.857,3.573) | 0.079(0.034,0.184)* | 0.001 |
| knowledge | | | | | |
| poor | 82(90.1) | 459(68.2) | 1 | 1 | |
| good | 9(9.9) | 214(31.8) | 4.248(2.095,8.615) | 4.555(2.130,9.740)* | 0.001 |

Where 1 = Reference,* shows the variable significant at p-value < 0.05 in multi variable analysis.

Masks can be used for the protection of healthy people to protect themselves when in touch with an infected person or for source control to prevent onward transmission by an infected person or both [7].

According to the findings, the overall knowledge of the students was 223 (29.2%), their attitude was 673 (88.1%), and their practice was 684 (89.5%), When compared to a previous study conducted in a healthcare worker's on proper face mask utilization and associated factors in police health facilities in Addis Ababa, Ethiopia [5], this was found to be 1.2 times lower for knowledge, two times higher for attitude, and 2.5 times higher for practice.

This study showed that students' overall utilization of face masks was more than 1.5 times greater than a survey conducted in Malaysia [2, 5], which is similar to the reported from Hong Kong Chinese study [20] and the reported in a Saudi Arabian study [21]. A systematic review and meta-analysis undertaken to investigate the efficiency of face masks in preventing respiratory virus transmission revealed that proper mask use by healthcare and non-healthcare

**Table 7. Factors affecting practice of the respondents on knowledge, attitudes, and practices of face mask utilization and associated factors in COVID-19 pandemic among Wachemo university students, southern Ethiopia: February to March 2021 (n = 764).**

| Variables | Practice Of Face Mask | | | | |
|---|---|---|---|---|---|
| | Poor | Good | COR(95%CI) | AOR(95%CI) | P-value |
| College | | | | | |
| Medicine and health sciences | 21(26.2) | 316(46.2) | 1 | 1 | |
| Natural and computational sciences | 50(62.5) | 169(24.7) | 0.225(0.131,0.357) | 0.23(0.13,0.40)* | 0.001 |
| Social sciences and law | 9(11.2) | 199(29.1) | 1.469(0.660,3.273) | 1.94(0.86,4.34) | 0.109 |
| Knowledge | | | | | |
| Poor | 71(88.8) | 470(68.7) | 1 | 1 | |
| Good | 9(11.2) | 214(31.3) | 3.592(1.762,7.321) | 4.40(2.13,9.14)* | 0.001 |

Where 1 = Reference,* shows the variable significant at p-value < 0.05 in multi variable analysis.

workers can lower the risk of respiratory virus infection by 80% [22]. The current study found that 89.5% of people in the study area used face masks, which is about 2.5 times higher than the research done in Ethiopian police health facilities [5]. This is almost 2.5 times higher than a study conducted at Dessie referral hospital in Northeast Ethiopia [23] and a study conducted in Pakistan [24].

The current study was in line with a study conducted in Hong Kong among community adults, in which the respondents indicated that they always wore face masks when caring for family members who had fevers or respiratory infections [25].

In terms of health professionals' attitudes regarding face mask use, the current study found 2 times as many positive attitudes toward correct face mask use in police health facilities in Addis Ababa, Ethiopia [5]. These discrepancies could be explained by a qualitative study conducted among Vietnamese healthcare workers, which found that most discussants were hesitant to use face masks to protect against respiratory disease because they lacked adequate data on the effectiveness of face masks in the prevention of respiratory illnesses such as COVID-19 and other commentators also bring up the possibility that wearing a face mask could hurt the patient's feelings [6].

According to the present study, 29.2% of students had a good understanding of how to use face masks. The result was 2.5 times lower than a study conducted in Ethiopia's Addis Abebe police health facilities and 3 times lower than a study conducted in Dessie, Ethiopia [1, 23] and 2.5 times lower than in research done in Public Malaysia [2].

When compared to students in college medicine and health sciences, students in college natural and computational sciences were 76.8% less likely to be associated with poor practices in the use of face masks during the COVID -19 pandemic (AOR: 0.23; 95% CI: 0.13, 0.40). This could be because different colleges in higher education had varied grades, understandings, and subject areas.

Students with good knowledge were 4.4 times more likely than students with poor knowledge to have poor practices when using face masks (AOR = 4.40; 95 percent CI; 2.13, 9.14). This could be due to a lack of understanding, and those students with good knowledge would try to use the face mask properly. The diverse study demographics, sample size determination, and operational definition could all play a role in this disparity.

## 5. Conclusion

When compared to other research, WCU students had positive attitudes and good practices regarding the use of face masks, but they had little knowledge about the use of face masks. Face mask use was strongly linked to the college of natural and computational science and having a thorough understanding of face mask use. For those at risk of a COVID-19 pandemic, comprehensive awareness efforts and education initiatives on the use of about-face masks by authorities were required. During this epidemic, authorities should develop policies and guidelines that cover the many types of face masks, their use, and the importance of face mask utilization.

### 5.1 Limitation

There are some drawbacks to this study. One of the drawbacks is bias as a result of the study design (cross-sectional) because the data were collected at specific time points and cause and effect relationships could not be investigated. This research was also restricted to a quantitative method. Furthermore, a lack of comparable research hampered the ability to compare the findings of this study to those of other investigations.

## Supporting information

**S1 File.**
(RAR)

## Acknowledgments

Special thanks are extended to local health managers, data collectors, supervisors, and Wachemo University.

## Author Contributions

**Conceptualization:** Yilma Markos Larebo, Desta Erkalo Abame.

**Data curation:** Yilma Markos Larebo, Desta Erkalo Abame.

**Formal analysis:** Yilma Markos Larebo, Desta Erkalo Abame.

**Funding acquisition:** Yilma Markos Larebo, Desta Erkalo Abame.

**Investigation:** Yilma Markos Larebo, Desta Erkalo Abame.

**Methodology:** Yilma Markos Larebo, Desta Erkalo Abame.

**Project administration:** Yilma Markos Larebo, Desta Erkalo Abame.

**Resources:** Yilma Markos Larebo.

**Software:** Yilma Markos Larebo, Desta Erkalo Abame.

**Supervision:** Yilma Markos Larebo, Desta Erkalo Abame.

**Validation:** Yilma Markos Larebo, Desta Erkalo Abame.

**Visualization:** Yilma Markos Larebo, Desta Erkalo Abame.

**Writing – original draft:** Yilma Markos Larebo, Desta Erkalo Abame.

**Writing – review & editing:** Yilma Markos Larebo, Desta Erkalo Abame.

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
