## [Decision Letter · Decision Letter 0]

10 May 2021

PONE-D-21-11089

Knowledge, Attitudes and Practices of Face Mask Utilization and Associated Factors in Covid -19 Pandemic Among Wachemo University Students, Southern Nation Nationality People Region: A Cross-Sectional Study

PLOS ONE

Dear Dr. larebo,

Thank you for submitting your manuscript to PLOS ONE. After careful consideration, we feel that it has merit but does not fully meet PLOS ONE’s publication criteria as it currently stands. Therefore, we invite you to submit a revised version of the manuscript that addresses the points raised during the review process.

We look forward to receiving your revised manuscript.

Kind regards,

Jenny Wilkinson, PhD

Academic Editor

PLOS ONE

Journal Requirements:

4. We suggest you thoroughly copyedit your manuscript for language usage, spelling, and grammar. If you do not know anyone who can help you do this, you may wish to consider employing a professional scientific editing service.  

5. Please include additional information regarding the validation and development of the questionnaire.

6. We note that you have indicated that data from this study are available upon request. PLOS only allows data to be available upon request if there are legal or ethical restrictions on sharing data publicly. For information on unacceptable data access restrictions, please see http://journals.plos.org/plosone/s/data-availability#loc-unacceptable-data-access-restrictions.

7. Your ethics statement should only appear in the Methods section of your manuscript. If your ethics statement is written in any section besides the Methods, please move it to the Methods section and delete it from any other section. Please ensure that your ethics statement is included in your manuscript, as the ethics statement entered into the online submission form will not be published alongside your manuscript.

Reviewers' comments:

Reviewer's Responses to Questions

**Comments to the Author**

1. Is the manuscript technically sound, and do the data support the conclusions?

Reviewer #1: Partly

Reviewer #2: Partly

Reviewer #3: Partly

2. Has the statistical analysis been performed appropriately and rigorously? 

Reviewer #1: Yes

Reviewer #2: No

Reviewer #3: Yes

3. Have the authors made all data underlying the findings in their manuscript fully available?

Reviewer #1: Yes

Reviewer #2: Yes

Reviewer #3: No

4. Is the manuscript presented in an intelligible fashion and written in standard English?

Reviewer #1: No

Reviewer #2: No

Reviewer #3: No

5. Review Comments to the Author

Reviewer #1: The authors presented a very important topic that if properly addressed would help control the spread of COVID-19 pandemic. They intended to identify the KAP of college students as regards the use of face masks.

major changes are needed to make this manuscript suitable for publication:

Introduction section is very long, not properly constructed, no flow of ideas and lacks study rationale. Authors may start by describe the types , importance and uses of face masks mainly during the current pandemic, then present the current evidence in this both internationally and in their country, with a proper rationale statement/paragraph.

the sentences in lines 193, 195, 196 need to be corrected to avoid misunderstanding.

methods section: The results of the students tested in the pilot study were included in the study?

How did the authors use the dependent variable?

all questionnaire items need meticulous language revision and proof reading:

Many questionnaire items are not suitable for the subtitled domain, hence doesn't reflect the targeted domain, e.g.,

Among the Attitude items 27,28,29,30,31,32,34 don't fit as attitude testing items

For practice section : 58,59,60,61,63,64,65 test knowledge or attitude but not practice.

Items with repeated meaning in the questionnaire as

56, 57

63,64,65

many Items are difficult to be understood e.g., 19, 40

results section and discussion need to revised after correcting the methods

conclusion does not reflect the results: awareness campaigns, educational session are needed not training sessions (results of knowledge is poorer than practice and attitude , so training will not solve the problem

Reviewer #2: This is an interesting study of the how facemasks are viewed in the study population during the COVID-19 pandemic. However, there are a few concerns that warrant a revision of the manuscript prior to publication.

1. The introduction is overly long and contains contradictory information. For example, the WHO stance is mentioned to be both not recommending and recommending masks. The paragraph starting on line 55 seems out of place. A major edit of this section would greatly benefit readability and help emphasize the main point of the paper.

2. The methods regarding how the study population was selected is unclear. Specifically, if the university has approximately twice as many male students, why is the study population primarily female?

3. The survey questions themselves, and how they were scored, are unclear. Were they provided to the participants exactly as shown in the appendix? How were the questions developed?

4. The text of the results section repeats a lot of what is in the tables and figures, and there are typographical errors.

5. The comparison with other similar studies is a bit of an overreach, as the methods varied. However, the authors' conclusions of their own study was reasonable given the study methods.

Overall, this manuscript has the potential to share some interesting information, and may prove especially valuable to locations similar to the study location. However, the manuscript is not publication ready as-is and consultation with a skilled medical editor is suggested.

Reviewer #3: This study examines the knowledge, attitudes and practices surrounding the use of face masks. The topic explored is an important one, especially given its use as a public health measure across the world and the controversy associated with it. While this study attempts to answer an important question, I have a few issues regarding it.

-Abstract: Only one set of results are presented in the abstract. Was this the primary outcome?

-Introduction: Please provide references for the statements made in the introduction line 85, line 88, line 110.

-Line 115: what is the difference between pre-asymptomatic and asymptomatic?

Independent variables: lines 193-198, I assume these are the independent variables but they need to be phrased as such.

-Please provide the objectives and the primary/secondary outcomes of the study.

- What was the rationale behind collecting data on religion, marital status, living situation etc. Did the authors have a hypothesis on whether these variables would affect the outcome?

- Discussion: I'd like to see some explanation and discussion of the results, in addition to comparison with other studies. Give potential explanations as to why there were differences between the groups. e.g. why do the authors think that those who were married had higher rates of negative attitudes?

- Would there have been an element of social desirability bias when questioning students on their attitudes towards face masks?

- Please describe the limitations of the study

-Conclusion: The authors state that "comprehensive training about face mask utilization should be designed and given by authorities". This needs to be expanded in the discussion. How did this study show that training by authorities was required? The study population was quite niche (young students, high education background), these results may not be generalizable to the whole population, so conclusions need to be made with this caveat in mind.

- The manuscript would benefit from a review of the English language/grammar, with particular attention given to tense, punctuation and sentence structure.

6. PLOS authors have the option to publish the peer review history of their article (what does this mean?). If published, this will include your full peer review and any attached files.

Reviewer #1: No

Reviewer #2: No

Reviewer #3: No

---

## [Author Response · Author response to Decision Letter 0]

23 Jun 2021

The authors do not have full mandate to share the data since they are the property of the funding institution.

I am not sole author 

the authors are Yilma Markos Larebo and Desta Erkalo Abame

All authors declared that there is no conflict of interest and proved final research manuscript.

---

## [Editor Report · Decision Letter 1]

25 Jun 2021

PONE-D-21-11089R1

Knowledge, Attitudes, and Practices of Face Mask Utilization and Associated Factors in COVID -19 Pandemic Among Wachemo University Students, Southern Nation Nationality People Region: A Cross-Sectional Study

PLOS ONE

Dear Dr. Larebo,

Thank you for submitting your manuscript to PLOS ONE. After careful consideration, we feel that it has merit but does not fully meet PLOS ONE’s publication criteria as it currently stands. Therefore, we invite you to submit a revised version of the manuscript that addresses the points raised during the review process.

We look forward to receiving your revised manuscript.

Kind regards,

Jenny Wilkinson, PhD

Academic Editor

PLOS ONE

Additional Editor Comments (if provided):

Thank you for your response. It is not clear from your response what your response is to each of the comments made by the reviewers and then what changes, if any, have been made to the manuscript as there are no track changes showing in submitted work.

Please provide a response to each of the reviewer comments, you may wish to use a table or different format (e.g. bolding reviewer comment/ plain text for your response) to make it clear what your response is and what changes you have made to the manuscript. Please also submit the revised manuscript with track changes.

---

## [Author Response · Author response to Decision Letter 1]

1 Jul 2021

Thank you!!!!

yes you are write and we tried to address the comment given to us and the ethical statement which is written in the methods part and online summation form is similar

---

## [Decision Letter · Decision Letter 2]

21 Jul 2021

PONE-D-21-11089R2

Knowledge, Attitudes, and Practices of Face Mask Utilization and Associated Factors in COVID -19 Pandemic Among Wachemo University Students, Southern Nation Nationality People Region: A Cross-Sectional Study

PLOS ONE

Dear Dr. Larebo,

Thank you for submitting your manuscript to PLOS ONE. After careful consideration, we feel that it has merit but does not fully meet PLOS ONE’s publication criteria as it currently stands. Therefore, we invite you to submit a revised version of the manuscript that addresses the points raised during the review process.

We look forward to receiving your revised manuscript.

Kind regards,

Jenny Wilkinson, PhD

Academic Editor

PLOS ONE

Additional Editor Comments:

Thank you for your responses. The responses to the reviewers’ comments are in many cases unclear and I recommend that you review these original comments and ensure that they are addressed, or a detailed response as to why changes have not been made are provided. The items below are particularly of note:

1. Reviewer comments relating to the methods. Specifically,

a. How were the questions developed and validated including how they were assigned to domains? This information needs to be included in the manuscript.

b. How were the questions scored?

c. What was the rationale for the way in which the attitude questions were scored & why was being above the average considered a positive response particularly give this is nonparametric data (line 155-159). Please provide the statistical rationale for this analytic decision.

d. Was the questionnaire provided exactly as in the appendix?

e. If the student population is predominantly male yet the study population is mainly female then this suggests bias in the study recruitment process and limits generalisability of the results. This is not addressed in the manuscript

2. The English has improved but there remain sentences that are unclear, for example “The study showed that the overall knowledge, attitude, and practice of the students towards face 37 mask utilization were 223 (29.2%), 673 (88.1%), and 684 (89.5%) respectively” – it is unclear what is being reported here (i.e. 29.2% for knowledge actually mean). Further it is incorrect to say that a value is ‘about’ and then give the exact value (e.g. line 196 “About 291(38.1%) ..”). Please carefully edit the work for English grammar and spelling.

3. Line 57 “COVID-19 is a virus …” is not correct. The name of the virus is severe acute respiratory syndrome coronavirus 2 (SARS-CoV-2).

4. Please provide the objectives and the primary/secondary outcomes of the study

5. For the cumulative GPA to be meaningful to readers the scale for this measure needs to be given e.g. 1 to 7 with a pass at 4. Please use the same abbreviation for this throughout. What was the rationale for use of 3.25 as threshold for categorising students into two groups?

Reviewers' comments:

Reviewer's Responses to Questions

**Comments to the Author**

1. If the authors have adequately addressed your comments raised in a previous round of review and you feel that this manuscript is now acceptable for publication, you may indicate that here to bypass the “Comments to the Author” section, enter your conflict of interest statement in the “Confidential to Editor” section, and submit your "Accept" recommendation.

Reviewer #1: (No Response)

Reviewer #3: All comments have been addressed

2. Is the manuscript technically sound, and do the data support the conclusions?

Reviewer #1: No

Reviewer #3: Yes

3. Has the statistical analysis been performed appropriately and rigorously? 

Reviewer #1: Yes

Reviewer #3: I Don't Know

4. Have the authors made all data underlying the findings in their manuscript fully available?

Reviewer #1: Yes

Reviewer #3: Yes

5. Is the manuscript presented in an intelligible fashion and written in standard English?

Reviewer #1: No

Reviewer #3: No

6. Review Comments to the Author

Reviewer #1: Thanks to the respected authors for their effort in revising the manuscript.

Unfortunately, many responses were not satisfying and did not answering the raised queries; mainly the points related to questionnaire items and conclusion.

Check the typo mistake in dependent variables sentence in methods section and correct it. Now all variables are dependent.

Reviewer #3: Thank you for revising the manuscript. All comments have been addressed well but I think this article would benefit from a language review prior to publication.

7. PLOS authors have the option to publish the peer review history of their article (what does this mean?). If published, this will include your full peer review and any attached files.

Reviewer #1: No

Reviewer #3: No

---

## [Editor Report · Decision Letter 3]

11 Aug 2021

PONE-D-21-11089R3

Knowledge, Attitudes, and Practices of Face Mask Utilization and Associated Factors in COVID -19 Pandemic Among Wachemo University Students, Southern Nation Nationality People Region: A Cross-Sectional Study

PLOS ONE

Dear Dr. Larebo,

Thank you for submitting your manuscript to PLOS ONE. After careful consideration, we feel that it has merit but does not fully meet PLOS ONE’s publication criteria as it currently stands. Therefore, we invite you to submit a revised version of the manuscript that addresses the points raised during the review process.

We look forward to receiving your revised manuscript.

Kind regards,

Jenny Wilkinson, PhD

Academic Editor

PLOS ONE

Additional Editor Comments:

Please provide an itemised response to each reviewer comment explaining your response and any revisions you have made to the manuscript. Where you have chosen not to make a suggested change please explain your reason for this.
---

## [Author Response · Author response to Decision Letter 3]

25 Aug 2021

Thank you for all!!!

We tried to address the comments which is forwarded to us and if any new we tried to hear from your side.

---

## [Editor Report · Decision Letter 4]

31 Aug 2021

PONE-D-21-11089R4

Knowledge, Attitudes, and Practices of Face Mask Utilization and Associated Factors in COVID-19 Pandemic Among Wachemo University Students, Southern Ethiopia : A Cross-Sectional Study

PLOS ONE

Dear Dr. Laredo,

Thank you for submitting your manuscript to PLOS ONE. After careful consideration, we feel that it has merit but does not fully meet PLOS ONE’s publication criteria as it currently stands. Therefore, we invite you to submit a revised version of the manuscript that addresses the points raised during the review process.

We look forward to receiving your revised manuscript.

Kind regards,

Jenny Wilkinson, PhD

Academic Editor

PLOS ONE

Journal Requirements:

Additional Editor Comments:

Thank you for your revisions. Some further minor revisions are required:

1. Please number tables and figures in the order in which they appear e.g. Table 1, Table 2, Table 3 etc rather than having Table 1, Table 2a, Table 3a, Table 2b etc

2. All references in the Reference list need to be checked that they contain all relevant information e.g. URL provided for websites, correct format, journal title provided etc

3. There are a few grammatical and punctuation errors that can be eliminated by careful proofreading
---

## [Editor Report · Decision Letter 5]

6 Sep 2021

Knowledge, Attitudes, and Practices of Face Mask Utilization and Associated Factors in COVID-19 Pandemic Among Wachemo University Students, Southern Ethiopia : A Cross-Sectional Study

PONE-D-21-11089R5

Dear Dr. Laredo,

We’re pleased to inform you that your manuscript has been judged scientifically suitable for publication and will be formally accepted for publication once it meets all outstanding technical requirements.

Kind regards,

Jenny Wilkinson, PhD

Academic Editor

PLOS ONE

Additional Editor Comments (optional):

Thank you for your revisions
---

## [Editor Report · Acceptance letter]

10 Sep 2021

PONE-D-21-11089R5 

Knowledge, Attitudes, and Practices of Face Mask Utilization and Associated Factors in COVID-19 Pandemic Among Wachemo University Students, Southern Ethiopia: A Cross-Sectional Study 

Dear Dr. Larebo:

I'm pleased to inform you that your manuscript has been deemed suitable for publication in PLOS ONE. Congratulations! Your manuscript is now with our production department. 

Kind regards, 

on behalf of

Dr Jenny Wilkinson 

Academic Editor

PLOS ONE